# Ventilator-Associated Pneumonia in Immunosuppressed Patients

**DOI:** 10.3390/antibiotics12020413

**Published:** 2023-02-20

**Authors:** Louis Kreitmann, Alexandre Gaudet, Saad Nseir

**Affiliations:** 1Médecine Intensive Réanimation, CHU de Lille, F-59000 Lille, France; 2Centre for Antimicrobial Optimisation, Department of Infectious Disease, Faculty of Medicine, Imperial College London, London W12 0HS, UK; 3ICU West, The Hammersmith Hospital, Du Cane Road, London W12 0HS, UK; 4Université de Lille, CNRS, Inserm, CHU Lille, Institut Pasteur de Lille, U1019-UMR9017-CIIL-Centre d’Infection et d’Immunité de Lille, F-59000 Lille, France; 5Inserm U1285, Université de Lille, CNRS, UMR 8576-UGSF, F-59000 Lille, France

**Keywords:** ventilator-associated pneumonia, immunosuppression, multidrug resistance, critical illness

## Abstract

Immunocompromised patients—including patients with cancer, hematological malignancies, solid organ transplants and individuals receiving immunosuppressive therapies for autoimmune diseases—account for an increasing proportion of critically-ill patients. While their prognosis has improved markedly in the last decades, they remain at increased risk of healthcare- and intensive care unit (ICU)-acquired infections. The most frequent of these are ventilator-associated lower respiratory tract infections (VA-LTRI), which include ventilator-associated pneumonia (VAP) and tracheobronchitis (VAT). Recent studies have shed light on some of the specific features of VAP and VAT in immunocompromised patients, which is the subject of this narrative review. Contrary to previous belief, the incidence of VAP and VAT might actually be lower in immunocompromised than non-immunocompromised patients. Further, the relationship between immunosuppression and the incidence of VAP and VAT related to multidrug-resistant (MDR) bacteria has also been challenged recently. Etiological diagnosis is essential to select the most appropriate treatment, and the role of invasive sampling, specifically bronchoscopy with bronchoalveolar lavage, as well as new molecular syndromic diagnostic tools will be discussed. While bacteria—especially gram negative bacteria—are the most commonly isolated pathogens in VAP and VAT, several opportunistic pathogens are a special concern among immunocompromised patients, and must be included in the diagnostic workup. Finally, the impact of immunosuppression on VAP and VAT outcomes will be examined in view of recent papers using improved statistical methodologies and treatment options—more specifically empirical antibiotic regimens—will be discussed in light of recent findings on the epidemiology of MDR bacteria in this population.

## 1. Introduction

Although the proportion of critically ill patients with immunosuppression has increased during the last decades, the prognosis of these patients has constantly improved. A recent large meta-analysis used individual patient data of studies that focused on immunocompromised patients with acute respiratory failure requiring invasive mechanical ventilation (IMV; 11,087 patients from 24 studies) [1]. The crude mortality rate was 53.2%, and adjusted survival improved over time (from 1995 to 2017, odds ratio [OR] for hospital mortality per year 0.96, 95% CI 0.95–0.97).

Immunosuppression was reported to be associated with higher rates of infection, related to reduced defense mechanisms and higher exposure to healthcare facilities, and antimicrobial courses [2]. However, few studies have carefully examined the relationship between immunosuppression and ventilator-associated lower respiratory tract infection (VA-LRTI). The aim of this narrative review is to discuss recent data on this issue and to determine further required research in this field (Figure 1).

## 2. Methods

We searched the MEDLINE and Pubmed databases with the following search query (using MeSH terms): “Ventilator-associated pneumonia AND (Immunocompromised Host OR leukopenia OR neoplasms OR HIV OR transplants OR Autoimmune Diseases OR steroids OR Immunosuppressive Agents)”. Further references were added through hand-searching in the relevant literature and verifying references of key papers. We screened titles and abstracts of papers identified by our search, and assessed the full text of potentially relevant articles. The inclusion of papers in the final manuscript was based on consensus among all three coauthors.

## 3. Incidence of VAP in Immunosuppressed Patients

Based on the data from the international prospective TAVeM database, Moreau et al. estimated the global incidence of VA-LRTI to be 16.6% in immunocompromised patients [3]. Interestingly, the authors reported a lower incidence of ventilator-associated tracheobronchitis (VAT) (7% vs. 12%) and pneumonia (VAP) (9% vs. 13%) in patients with immunosuppression compared with those with no immunosuppression, with a VAP incidence similar to those previously described in the lung and liver transplant populations [4,5]. In addition, VAP was more frequently caused by multidrug-resistant (MDR) bacteria in immunocompromised patients, compared with those with no immunosuppression (78% vs. 58%). One of the hypotheses to explain these results was a higher frequency of antibiotic exposure in immunocompromised patients, leading on the one hand, to a reduced risk of VA-LRTI, and on the other hand, to the emergence of resistant mutants. Conversely, the results of a retrospective multicenter study suggest that the initiation of corticosteroid treatment in the initial phase of COVID-19 would be associated with a higher incidence of VAP, estimated at 63% vs. 49% in patients not receiving corticosteroids [6]. These latter data seemed consistent with the results from other previously published smaller series [7], contrasting with the data from the TAVeM database, which mostly depicted patients with prior immunosuppression. However, duration of corticosteroid treatment was not taken into account in these studies, and whether these patients could be considered as immunosuppressed, because of corticosteroid treatment is unclear.

## 4. Diagnosis

The diagnosis of VAP often remains tricky in immunocompromised patients. First of all, it must be reminded that the diagnosis of VAP is based on symptoms of lower respiratory tract infection in patients who have been intubated for over 48 h and have a positive result from a lower respiratory microbiological sample, along with the detection of a new infiltrate on chest radiography. Moreover, the diagnosis of VAT is based on the combination of the previously mentioned criteria, without any new radiographic infiltrates [8]. 

The first challenge lies in the frequency of non-infectious differential diagnoses, which may be difficult to distinguish from VAP, due to atypical presentations in immunocompromised patients, including the absence of an inflammatory syndrome. These differential diagnoses include pulmonary toxicities of oncological treatments, acute pulmonary edema, intra-alveolar hemorrhage, or lesions related to the underlying disease itself. The frequent cohabitation of these entities to varying degrees with a genuine infectious process makes the CT scan one of the first-line imaging exams in the case of VAP in the immunocompromised [9]. The CT scan makes it possible to describe certain aspects orienting the infectious etiological diagnosis. Thus, consolidations are frequently associated with bacterial etiologies, whereas ground-glass opacities are mostly found in viral pneumonia [10,11]. Other strategies to document differential diagnoses, such as cardiac and pleural ultrasound, have also emerged as primary diagnostic tools in this population. 

The use of biomarker tests, such as PCT or CRP for monitoring VAP has been suggested, although these studies have not been specifically conducted in immunocompromised individuals. Luyt et al. found that PCT had a low accuracy for diagnosing VAP, with areas under the ROC curves at 0.51 and 0.62 for D-1 PCT and PCT increase, respectively [12]. In the BioVAP study, Povoa et al. reported that CRP had a moderate performance in diagnosing VAP, with areas under the ROC curves at 0.7 and 0.8 for the CRP variations [13]. Interestingly, the authors also observed that a single measurement of CRP could be useful for excluding a VAP diagnosis.

Microbiological documentation is the cornerstone of the diagnosis of VAP in the immunocompromised. This can be all the more delicate as fungal, in particular Aspergillus, and viral etiologies, mostly herpes viruses, can frequently be added to the bacterial infectious agents usually encountered. It must be first noted that the type of immunosuppression is an important element to take into account in the etiological assessment. Thus, T cell deficiencies, hematopoietic stem cell transplantation (HSCT) and high dose steroid treatments are well known risk factors for herpes virus pneumonia [14]. The same risk factors have been described for Aspergillus, along with severe neutropenia, acute myeloid leukemia, and massive environmental exposures [15]. However, these risk factors remain inconstantly associated with the etiological diagnosis. Fiberoptic bronchoscopy (FOB) with bronchoalveolar lavage (BAL) remains to this day one of the most exhaustive techniques for exploring these different possibilities. Previously published data reported similar performances between BAL and non-invasive tests in identifying the cause of acute respiratory failure in a community setting in immunocompromised patients [16]. Indeed, in this study, BAL and non-invasive strategies conducted to similar diagnosis rates, found at 79.6% and 78.3%, respectively. However, several elements underline the interest of BAL and should be taken into account in the choice of the examination in immunocompromised patients. The interest of this technique lies first of all in the possibility of making macroscopic findings, such as the herpetic lesions observed in HSV pneumonia [17]. BAL is also the reference technique for viral pneumonia usually encountered in intubated patients, for which the diagnosis is based on PCR, with the possible presence of intranuclear inclusion and detection of giant cells in the case of HSV pneumonia. Thus, high accuracies of PCR performed in BAL have been reported, with areas under the ROC curves found at 0.89 and 0.91, respectively, for the diagnosis of HSV and CMV pneumonia [17,18,19,20]. In addition, BAL is also of particular interest for the diagnosis of invasive pulmonary aspergillosis (IPA), for which recent data suggest a high frequency of about 12% in patients with suspected VAP [21]. Direct detection of Aspergillus in culture on BAL has a high specificity, but a low sensitivity, ranging from 30 to 60% [22], with a delay of 2 to 5 days. The detection of galactomannan antigen and the Aspergillus PCR on BAL allows for a rapid diagnosis, delivered within one day. Repeated Aspergillus PCR in BAL yields a high specificity above 0.90 for the detection of IPA, but remains associated with more variable sensitivities, sometimes found below 0.80 [22]. Moreover, the sensitivity and specificity of BAL galactomannan with an index cutoff of 1.0 are constantly found above 0.90. These exams therefore are among the first-line tests in immunocompromised patients, at risk of IPA. Serum tests, based on the detection of galactomannan, β-D-glucans and Aspergillus PCR, are more accessible, but have a lower performance for the diagnosis of IPA [22]. Indeed, Aspergillus PCR yields a low sensitivity, yet with high specificity for the diagnosis of IPA. In addition, serum galactomannan and β-D-glucans both have low accuracies for the diagnosis of IPA, with sensitivity and specificity found below 0.90. These non-invasive tests are therefore to be considered as surveillance methods, which should lead to the performance of BAL in the case of a suspected diagnosis. 

If the interest of PCR seems to be well established in the context of viral and fungal pneumonia, several characteristics of this method make it a technique of interest in the context of bacterial VAP in the immunocompromised. Indeed, published data suggest that the value of PCR compared to conventional culture in the correct identification of the bacterial etiology of VAP [23], particularly in patients previously exposed to antibiotics, is a situation that seems to be more frequently encountered in the immunocompromised [3]. Thus, among patients who had received prior antibiotics, Strålin et al. reported that BAL culture was positive in 5/24 (21%) cases, while this rate reached 14/24 (58%) cases for PCR [23]. The interest of PCR also lies in its capacity to provide a rapid etiological diagnosis, of the order of a few hours vs. a few days for culture [24], coupled with the possibility of detecting the presence of MDR [25]. These characteristics also seem to be of crucial importance, given the increased frequency of the involvement of MDR bacteria in VAP in the immunocompromised [3]. The potential value of PCR was highlighted in a two-center study that included a majority of immunocompromised patients, in which a strategy based on multiplex PCR on BAL specimens appeared to reduce inappropriate treatment time [26]. 

Among the more recently developed techniques, metagenomic next-generation sequencing (mNGS) is a high-throughput technique used for unplanned pathogen detection. The mNGS analysis on BAL seems to allow for a wide detection of viral, bacterial, and fungal infections, with a better performance than conventional culture in VAP [27]. Retrospective data suggest a gain in performance of mNGS over standard culture techniques particularly marked in immunocompromised patients, especially with a much higher sensitivity of mNGS for bacterial infections (0.90 vs. 0.50) and coinfections (0.69 vs. 0.48) [28]. These results underline the need for prospective evaluations to corroborate the benefit of this technique in this population.

The expected benefits of FOB with BAL must nevertheless be weighed in balance with the side effects of this technique. These are mainly represented by transient hypoxemia, pneumothorax, and bleeding. Hypoxemia is by far the most frequent complication, found in one in four patients immediately after BAL in an observational study of 164 mechanically ventilated intensive care patients [29]. Nevertheless, this study underlined the transient and mostly reversible character of hypoxemia, as well as the absence of bleeding or pneumothorax in this context. These results were consistent with previously published data, which emphasized the exceptional nature of these complications. Another study carried out in a population of onco-hematology patients also suggested the very low frequency of bleeding events after BAL, with an estimated occurrence of severe bleeding of 1/500, with no apparent impact on the severity of thrombocytopenia [30]. Another study published by Kamel et al. clarified these data [31]. In this study, the authors described the frequency of complications occurring within 24 h after BAL in a population of 378 ventilated patients, including nearly 50% of immunocompromised patients. The most frequently documented events within 24 h after BAL were arterial hypotension, found in 34.9% of cases, and a fall in SpO2, found in 12.2% of cases. The frequency of hemoptysis after BAL was 1.6%. Death within 24 h after BAL occurred in 3.2% of cases. It should be noted, however, that this study did not make it possible to establish the imputability of BAL in the events recorded afterwards, particularly because of the absence of control patients. It can also be noted that most of the events reported, including hypotension and desaturation, did not lead to a substantial modification of the administered therapies, underlining a moderate impact on patient management. The main category at risk of complication remained the group of patients on high-flow nasal oxygen or non-invasive ventilation, with a rate of intubation occurring within 24 h post-LBA of more than 16%, to be compared to the extremely low rate, below 1%, of life-threatening complications observed in intubated patients. These data suggest that the use of BAL remains mostly safe in ventilated patients, including immunocompromised patients, reinforcing the interest of this method for the diagnosis of VAP in this population [32].

## 5. Microbiology

Bacteria are the leading pathogens responsible for VAP and VAT in the general non-immunocompromised population, and this is also the case among immunocompromised patients [33,34,35]. Most cases (50–80%) of bacterial VALTRI are caused by gram negative bacteria, the most frequently isolated species being *Pseudomonas aeruginosa*, *Acinetobacter baumannii,* and *Enterobacteriaceae*, which include *Escherichia coli* and *Klebsiella pneumonia* [36,37]. Antibiotic resistance in gram negative bacteria has been steadily increasing worldwide and is of particular concern because of the paucity of new antimicrobials with activity against these pathogens [38]. Among *Enterobacteriaceae*, resistance to third- or fourth-generation cephalosporins due to the expression of extended-spectrum β-lactamases (ESBLs) and/or AmpC β-lactamases is an important worry, and so is the emergence of strains resistant to last resource antibiotics, such as carbapenems and colistin. The proportion of multidrug-resistant strains among nonfermenters (which in certain settings can be above ~30%) also imposes significant constraints in the choice of empirical antibiotic regimens [38]. In gram positive organisms, *Staphylococcus aureus* is the most frequently encountered species [39], but the diffusion of methicillin-resistant strains has been relatively controlled [40]. 

Both the bacterial species isolated in patients with VALTRI, and maybe more importantly their antibiotic susceptibility profile are influenced by several factors, such as hospital and ICU length-of-stay and IMV duration before VALTRI onset. Prior colonization and/or infection with MDR bacteria, prior to exposure to antibiotics, especially with broad-spectrum activity, local ecology (i.e., the site-specific background prevalence of MDR bacteria, as well as intra-hospital epidemics) are also important risk factors for MDR bacteria. Moreover, severity of acute illness, organ failures, shock, ARDS, and life support techniques have been repeatedly associated with VAP related to MDR bacteria, even though these variables act more likely as confounders than as direct causal factors [8,33,41,42,43]. Because of this, studying whether immunosuppression significantly and systematically modifies the distribution of bacterial species implicated in VALTRI could be distorted by a myriad of confounding factors, and would be of little overall significance. More interesting is the assessment of the impact of immunosuppression on the risk of colonization and infection, including VA-LTRI, with MDR bacteria. 

Numerous studies have investigated the prevalence of antimicrobial resistance (AMR) in immunocompromised patients (reviewed in [44,45]), and the dominant view in the literature has been that immunocompromised patients present a high risk of colonization and/or infection with MDR bacteria. In line with this, immunosuppression has been considered a risk factor for MDR bacteria in VAP [43], and guidelines have suggested to take this into consideration when selecting an empiric antibiotic regimen in immunocompromised patients with VAP [41,46]. However, most of the related studies present important limitations, including the lack of a formal comparison between immunocompromised and non-immunocompromised patients and the failure to take into account important patient-related confounding factors. In the CIMDREA study [47], an observational prospective multicenter cohort study in France, we found that the incidence of ICU-acquired colonization with MDR bacteria was lower in immunocompromised vs. non-immunocompromised patients (adjusted subhazard ratio [sHR] 0.56, 95% CI 0.4–0.79), but the incidence of ICU-acquired infection with MDR bacteria was not significantly different between groups (adjusted sHR 0.59, 95% CI 0.33–1.05). This was also true when focusing on VAP related to MDR bacteria (28-day cumulative incidence 33.3% and 38.3% in immunocompromised and non-immunocompromised patients, respectively). Importantly, one limitation of that study is that only the incidence of ICU-acquired infections related to MDR bacteria (i.e., not those related to sensitive strains) was investigated. 

In the study by Moreau et al., among patients with VALRTI, the rate of MDR bacteria was significantly higher among immunocompromised than among non-immunocompromised patients (72% vs. 59% of VALTRI episodes, OR 1.75, 95% CI 1.13–2.71) [3]. Similar results were found in the subgroup of patients with VAP (78% vs. 58% of VALTRI episodes, OR 2.39, 95% CI 1.29–4.48), but not in patients with VAT (65% vs. 60%, *p* = 0.52). However, as detailed above, because immunocompromised patients also had a lower cumulative incidence of VALTRI and VAP in comparison to their non-immunocompromised counterparts, the cumulative incidence of VALTRI and VAP related to MDR bacteria might in fact be comparable between groups (12.5% vs. 14.7% for VALTRI and 7.4% vs. 7.7% for VAP), which would be concordant with the results of the CIMDREA study [47]. Taken together, these studies suggest that critically-ill immunocompromised patients may not be at higher risk of being colonized and/or infected with MDR bacteria when accounting for key confounding factors, and that immunosuppression should not be considered separately from other risk factors when evaluating the need to prescribe broad-spectrum antibiotics empirically in the setting of VAP. 

In immunocompromised patients, more so than in their non-immunocompromised counterparts, VAP related to opportunistic fungal and viral pathogens are also an important concern. Regarding fungal infections, VAP related to *Aspergillus* sp. is the most pressing concern among immunocompromised patients, but its exact prevalence and epidemiological determinants are difficult to assess, mostly due to the absence of formal diagnostic criteria [22,48]. Immunosuppression is a well-established risk factor for community-onset invasive pulmonary aspergillosis (IPA), but it has been increasingly recognized that it could develop in mechanically ventilated patients, and thus lead to VAP, even more so when they are immunocompromised [49]. Classically, ICU-onset IPA has been described in patients receiving IMV in the setting of severe influenza [50,51], and in a retrospective multicenter cohort study on 432 critically-ill flu patients, its incidence was more than twice higher (32% vs. 14%) among immunocompromised than among non-immunocompromised patients [50]. Similarly, in a multicenter observational study on 563 ICU patients (of whom 86% were receiving IMV at the time of diagnosis), factors of immunosuppression were more prevalent among those with proven or putative IPA than among those with Aspergillus colonization [52]. VAP related to *Pneumocystis jirovecii* [53,54,55] (PCP) and *Cryptococcus* sp. [56,57] are uncommon in immunocompromised patients (especially for PCP in those receiving prophylaxis with cotrimoxazole), but both pathogens should be included in the diagnostic workup when no bacterial pathogen is identified. There is no data supporting the role of *Candida* sp. in VAP [58].

Finally, viruses have also been implicated in VAP, both in non-immunocompromised [59] and immunocompromised patients [60]. The pathophysiology of viral VAP involves either the reactivation of endogenous viruses from the *Herpesviridae* family, including *Herpes simplex virus* (HSV) and *Cytomegalovirus* (CMV), or more rarely ICU-acquired infections with respiratory pathogens (e.g., influenza, rhinovirus, etc.). Histologically-proven HSV [17] and CMV [61] pneumonia have been documented in ICU patients under IMV, and in a retrospective monocenter study on 710 ventilated patients, immunosuppression (adjusted OR 2.97, 95% CI 1.44–6.14) and stem-cell transplantation (aOR 3.58, 95% CI 1.17–10.99) were independently associated with viral replication in the lungs. However, large-scale data on the incidence and risk factors of viral VAP are limited, which can be related to the fact that detecting the genetic material of a virus, especially HSV or CMV, in the respiratory secretions of ventilated patients does not necessarily imply direct pathogenicity, which can only be proven through histopathology [61].

## 6. Impact of Immunosuppression on Outcomes in VAP and VAT Patients

Several studies have documented an increased mortality in unselected cohorts of critically-ill patients presenting with at least one episode of VAP during ICU stay, in comparison with patients with no VAP. In a meta-analysis of 24 trials (6284 patients), the overall attributable mortality of VAP was estimated at 13% [62]. However, the excess of mortality directly attributable to VAP has been a subject of debate, as it appears to be lower when estimated using statistical methodologies that appropriately account for competing risks and time-dependent confounding factors [63,64,65]. 

It has been suggested that the attributable mortality of VAP may be influenced by patient-related features (e.g., it may be higher in surgical than in medical patients) [62], but the effect of immunosuppression on the relationship between the occurrence of VAP and mortality has been scarcely investigated. In the study by Moreau et al. [3], which provides the most compelling evidence, both immunocompromised and non-immunocompromised patients with VAP had a higher mortality than patients with no VA-LRTI and patients with VAT, but more importantly, among patients with VALTRI and with VAP, ICU mortality was significantly higher in immunocompromised vs. non-immunocompromised patients (54% versus 30%, OR 2.68, 95% CI 1.78–4.02 in patients with VALTRI). Furthermore, among patients with VALTRI, immunosuppression was an independent predictor of ICU mortality (HR 1.6, 95% CI 1.19–2.16). In a meta-analysis of 26 studies published in 2010 on patients with microbiologically-confirmed VAP, malignancy was associated with an increased mortality (OR 2.20, 95% CI 1.10–4.40), but this estimate was not adjusted for confounders [66]. In the study on ICU-acquired IPA, bone-marrow transplant was an independent predictor of mortality (OR 3.352, 95% CI 1.060–10.598) [52]. Finally, studies with smaller sample sizes, a mono-centric design [67] and/or a focus on restricted sub-groups of immunocompromised patients—e.g., liver transplant [68] and HIV patients [69]—have not found a significant effect of immune status on the association between VALTRI or VAP and mortality, but firm conclusions cannot be drawn from these studies due to their methodological limitations.

VAT has not been associated directly with an increased mortality in ICU patients, but transition from VAT to VAP was found to be an independent risk factor for mortality in the TAVeM study [39]. Furthermore, appropriate antibiotic treatment of VAT has been shown to decrease the risk of transition from VAT to VAP [70], and has been associated with lower ICU mortality in the TAVeM study [39]. In the study by Moreau et al. on immunocompromised patients [3], the incidence of progression from VAT to VAP was not significantly different between immunocompromised and non-immunocompromised patients (13% vs. 12%, *p* = 0.69), suggesting that immunosuppression is not a risk factor for progression from VAT to VAP.

Both VAP [71] and VAT [39] have been associated with a higher duration of IMV and longer ICU length-of-stay in unselected cohorts of critically-ill patients. In the study by Moreau et al. [3], similar findings were obtained, but more importantly immunosuppression did not seem to modify the impact of VAP/VAT on these outcomes. As an example, in non-immunocompromised patients, median IMV duration was 14 days (IQR 8–26) in case of VAP and 13 days (8–22) in case of VAT, as compared to 7 days (4–12) in case of no VALTRI, and in immunocompromised patients it was 15 days (8–27) in case of VAP, 16 days (10–25.5) in case of VAT, as compared to 7 days (4–14) in case of no VALTRI. Similar trends were found when assessing the impact of immunosuppression on the association between VAP/VAT and ICU LOS.

Based on the available evidence, immunosuppression could amplify the impact of VAP occurrence on mortality, but more evidence on this is needed. Second, immunosuppression does not seem to modify the impact of VAT and VAP on ICU LOS and IMV duration, nor the probability of transitioning from VAT to VAP.

## 7. Treatment of VAP in Immunosuppressed Patients

Current European and American guidelines on VAP [8,42] recommend considering immunosuppression as a risk factor for MDR bacteria, and treating VAP in immunosuppressed patients with large spectrum combination therapy, including a betalactam with activity against *Pseudomonas aeruginosa*. However, these recommendations are based on common sense and on available evidence. In a single center observational study, our group found immunosuppression to be associated with increased risk for colonization or infection related to MDR bacteria [72]. However, after careful adjustment for antibiotic treatment and other potential confounders, immunosuppression was not independently associated with increased risk for colonization or infection related to MDR bacteria (ICU-MDR-col and ICU-MDR-inf, respectively). In a recent multicenter prospective observational study, our group evaluated the relationship between immunosuppression and ICU-MDR-col and/or ICU-MDR-inf in a cohort of 750 critically ill patients [47]. Following the adjustment for center and pre-specified baseline confounders, immunocompromised patients had a lower incidence rate of ICU-MDR-col and/or ICU-MDR-inf (adjusted incidence ratio 0.68, 95% CI 0.52–0.91). When considered separately, the difference was significant for ICU-MDR-col, but not for ICU-MDR-inf. These data suggest that large spectrum antimicrobial treatment in immunosuppressed patients with VAP might not be justified in stable patients with no other risk factors for MDR. Further large multicenter studies are needed to confirm these data before modifying the current recommendations.

Although no statement on antimicrobial treatment for VAT is provided in European recommendations, American guidelines recommend not treating VAT with antimicrobials (weak recommendation, based on low evidence level). To our knowledge, no specific data exist on the interest of treating VAT with antibiotics in immunosuppressed patients.

## 8. Conclusions

VAP and VAT are among the most frequent ICU-acquired infections in immunocompromised patients, but their incidence might actually be lower than in non-immunocompromised patients. While the proportion of VAP and VAT episodes related to MDR bacteria might be higher in immunocompromised patients (possibly owing to higher antibiotic exposure), recent data have shown that the cumulative incidence of ICU-acquired infections (including VAP) with MDR bacteria might be lower in immunocompromised patients than non-immunocompromised ones. The microbiology of VAP and VAT in this population includes bacteria (especially gram negative), fungi (notably *Aspergillus* sp.), and viruses. Accurate and timely detection of pathogens and their antibiotic resistance profile is key to delivering the optimal antibiotic treatment, which is associated with positive outcomes. The exact role of bronchoscopy and bronchoalveolar lavage, as well as recent multiplex PCR-based diagnostic tools and serum biomarkers warrants further investigation. Immunosuppression could amplify the impact of VAP occurrence on mortality. Finally, recent data on the epidemiology of AMR in critically-ill immunocompromised patients might help nuance guidelines on empirical antibiotic regimens in immunocompromised patients with VAP, but further studies are needed to complement these findings.

## Figures and Tables

**Figure 1 antibiotics-12-00413-f001:**
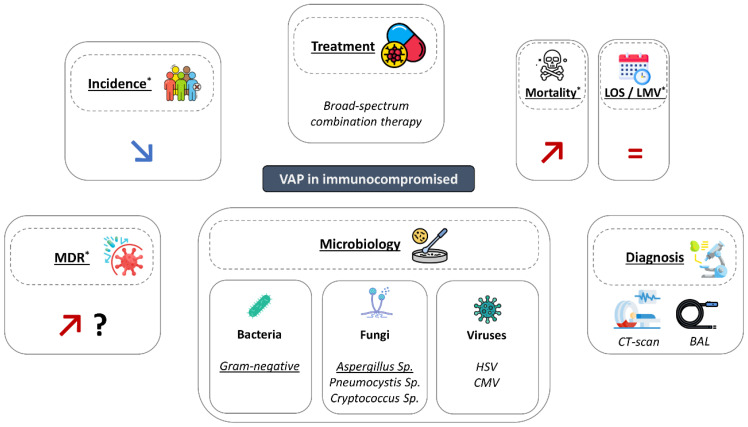
Main characteristics of ventilator-associated pneumonia in immunocompromised patients. BAL: broncho-alveolar lavage; LMV: length of mechanical ventilation; LOS: length of stay; MDR: multidrug resistant bacteria; VAP: ventilator-associated pneumonia. *: compared to immunocompetent patients.

## Data Availability

Not applicable.

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
