# Peer review of "Ventilator-Associated Pneumonia in Immunosuppressed Patients"

_antibiotics, 2023, doi:10.3390/antibiotics12020413_

Round 1

Author Response

Reviewer 1

Even though this is a narrative review, the author should explain how and why the article was included in this review (method).

Response. We have added a METHODS paragraph and the following description of our search and selection process: “We searched the MEDLINE and Pubmed databases with the following search query: “Ventilator-associated pneumonia” AND (Immunocompromised Host OR leukopenia OR neoplasms OR HIV OR transplants OR Autoimmune Diseases OR steroids OR Immunosuppressive Agents). Further references were added through hand searching of the relevant literature and verifying references of key papers on the subject. We screened titles and abstracts of papers identified by our search, and assessed the full text of potentially relevant articles. The relevance of papers to be included in the final manuscript was based on consensus among all three coauthors.” (lines 67-74).

In the abstract section, the author mentions some aspects, i.e. etiology, diagnostic, treatment, and impact. Therefore, the author should remain with that in order to keep consistency.

Response. We have re-written the abstract in order to follow the same order of paragraphs as in the text.

In the microbiology subsection: The author states that the etiology of infection is multidrug resistance, therefore, the author should add information on multidrug resistance evidence, the prevalence, and name of bacteriae, where and when (consider the selection pressure aspect).

Response. We thank the reviewer for this comment and have provided more details on mechanisms of resistance, prevalence and affected bacterial species on lines 196-203: “Among Enterobacteriaceae, resistance to third- or fourth-generation cephalosporins due to the expression of extended-spectrum β-lactamases (ESBLs) and/or AmpC β-lactamases is an important worry, and so is the emergence of strains resistant to last resource antibiotics such as carbapenems and colistin. The proportion of multidrug-resistant strains among nonfermenters (which in certain settings can be above ~30%) also imposes significant constraints in the choice of empirical antibiotic regimens[34]. In gram positive organisms, Staphylococcus aureus is the most frequently encountered species[35], but the diffusion of methicillin-resistant strains has been relatively controlled[36]”. We also consider the difference in selection pressure according to geographical settings on lines 208-209: (…) local ecology (i.e., the site-specific background prevalence of MDR bacteria, as well as intra-hospital epidemics)”

In the diagnosis subsection: The author should elaborate more on aspects of microbiology, i.e. laboratory examination, availability, accuracy, etc., because 'Microbiology is the cornerstone of the diagnosis'. How about another laboratory examination, such as procalcitonin and CRP? The author should reorder diagnostic tools from the most beneficial (the cornerstone) or their accuracy.

Response. We thank the reviewer for this comment. We added additional information about the diagnostic accuracies of microbiological lab tests throughout the diagnosis section. A short paragraph about the use of PCT and CRP as monitoring were added too. We propose not to reorder the content of the diagnosis section, as we aimed to deal with routinely used diagnostic tools first (viral PCR, fungal antigens and PCR…), and then address the question of bacterial PCR and mNGS which are not commonly used in the clinical practice.    

There are some factors affecting outcomes in VAP, why highlight immunosuppression?

Response. We thank the reviewer for this interesting remark. We have mainly focused on immunosuppression because it is the main topic of this review paper, and the subject of previous work by our team. The other main reason is that the literature on the impact of VAP on outcomes, especially its attributable mortality, is conflicting, and we reasoned that elaborating on the influence of patient-related factors on the relationship between occurrence of VAP and outcome would only be based on debatable findings from a limited pool of studies. However, we have modified lines 292-295 accordingly: “It has been suggested that the attributable mortality of VAP may be influenced by patient-related features (e.g., it may be higher in surgical than in medical patients)[62], but the effect of immunosuppression on the relationship between occurrence of VAP and mortality has been scarcely investigated.”

Diagnosis is associated with treatment, and treatment is associated with impact. The author should explain this in the conclusion section.

Response. We have added the following sentence to the conclusion: “Accurate and timely detection of pathogens and their antibiotic resistance is key to delivering the optimal antibiotic treatment, which is associated with positive outcomes.” (lines 367-369)

Reviewer 2 Report

The manuscript entitled Ventilator-associated pneumonia in immunosuppressed patients by Louis Kreitmann et al is very well written and a comprehensive review, I would suggest minor revisions.

The authors should address the following comments

Please add proper sub-section number in the revised manuscript.

If the author can add some figures in the revised manuscript it will be a good addition and will attract more attention from readers as the topic is very catchy and interesting.

Why the authors put figure 1 below the conclusion? 

Please add some latest references in the revised manuscript.

Author Response

The manuscript entitled Ventilator-associated pneumonia in immunosuppressed patients by Louis Kreitmann et al is very well written and a comprehensive review, I would suggest minor revisions. The authors should address the following comments.

Please add proper sub-section number in the revised manuscript.

Response. We have numbered each sub-section.

If the author can add some figures in the revised manuscript it will be a good addition and will attract more attention from readers as the topic is very catchy and interesting.

Response. We thank the reviewer for this suggestion. Our figure proposal was primarily intended to provide a concise overview of the points discussed in our review. In order to maintain the educational aspect of this approach, we propose to only keep one figure for this review.

Why the authors put figure 1 below the conclusion? 

Response. This is mainly an editing issue, as the Editor of the journal will be responsible for inserting the figure in the middle of the text.

Please add some latest references in the revised manuscript.

We thank the reviewer for this suggestion and have added some recent references (highlighted in blue) in the revised version of the manuscript.

Reviewer 3 Report

This is a paper on ventilator-associated pneumonia in immunosuppressed patients.

My only comment is that it's rather big and maybe, a bit tiring, towards the end.

Author Response

This is a paper on ventilator-associated pneumonia in immunosuppressed patients.

My only comment is that it's rather big and maybe, a bit tiring, towards the end.

Response. We thank the reviewer for this comment.

Reviewer 4 Report

The review organized by Louis Kreitmann et al. mainly focus on ventilator-associated pneumonia (VAP) and tracheobronchitis (VAT) in immunocompromised patients, sketch the specialties from different aspects include the incidence, diagnosis, microbiology composition and the effect in VAP and VAT patients that undergo with immunosuppression. They summarize that immunocompromised patients have a lower possibility to get the ICU-acquired infections with MDR bacteria. Bacteria, fungi and viruses especially gram-negative bacteria and Aspergillus sp. occupy a dominant ratio. Several methods and serum biomarkers could be used as diagnostic tools. While more efforts should be put on empirical antibiotic regimens in immunocompromised patients with VAP.

Minor points:

1. For diagnosis part, author give a clearly describe for VAP.  While for VAT, here is nothing mentioned. Even diagnose of VAT remain unclear, I think it is still necessary to present the current situation and possible diagnostic method. It would make people who don’t really know about VAT get a rough idea about VAT.

2. For Figure 1, please replace it with higher resolution.

3.Please check the spelling of the word. Page2, line 77, “weather” should be “whether”.

Author Response

The review organized by Louis Kreitmann et al. mainly focus on ventilator-associated pneumonia (VAP) and tracheobronchitis (VAT) in immunocompromised patients, sketch the specialties from different aspects include the incidence, diagnosis, microbiology composition and the effect in VAP and VAT patients that undergo with immunosuppression. They summarize that immunocompromised patients have a lower possibility to get the ICU-acquired infections with MDR bacteria. Bacteria, fungi and viruses especially gram-negative bacteria and Aspergillus sp. occupy a dominant ratio. Several methods and serum biomarkers could be used as diagnostic tools. While more efforts should be put on empirical antibiotic regimens in immunocompromised patients with VAP.

Minor points:

  1. For diagnosis part, author give a clearly describe for VAP.  While for VAT, here is nothing mentioned. Even diagnose of VAT remain unclear, I think it is still necessary to present the current situation and possible diagnostic method. It would make people who don’t really know about VAT get a rough idea about VAT.

Response. Following the reviewer’s comment, we added a paragraph to describe the respective definitions of VAP and VAT at the beginning of the diagnosis section.

  1. For Figure 1, please replace it with higher resolution.

Response. We thank the reviewer and have increased the resolution of Figure 1.

3.Please check the spelling of the word. Page2, line 77, “weather” should be “whether”.

Response. We thank the reviewer and have corrected this typo.